# Alkaloid Profile Characterisation and Bioactivity Evaluation of Bolivian *Hippeastrum* Species (Amaryllidaceae) as Cholinesterase Inhibitors

**DOI:** 10.3390/life15050719

**Published:** 2025-04-29

**Authors:** María Lenny Rodríguez-Escobar, Raúl Fernando Lara, Margoth Atahuachi, Alfredo F. Fuentes, Carla Maldonado, Jaume Bastida, Luciana R. Tallini, Laura Torras-Claveria

**Affiliations:** 1Departament de Biologia, Sanitat i Medi Ambient, Facultat de Farmàcia i Ciències de l’Alimentació, Universitat de Barcelona, Av. Joan XXIII n° 27-31, 08028 Barcelona, Spain; m-rodriguez@ub.edu (M.L.R.-E.);; 2Herbario Nacional Forestal Martin Cárdenas (BOLV), Centro de Biodiversidad y Genética, Facultad de Ciencias y Tecnología, Universidad Mayor de San Simón, Calle Sucre y Parque La Torre, Edificio Laboratorios, 4to piso, Cochabamba 4973, Bolivia; 3Herbario Nacional de Bolivia (LPB), Instituto de Ecología, Facultad de Ciencias Puras y Naturales, Universidad Mayor de San Andrés, Calle 27 y Andrȳs Bello s/n Cota Cota, Correo Central, La Paz 10077, Bolivia; alfrefuentes@gmail.com (A.F.F.);; 4Latin America Department, Missouri Botanical Garden, 4344 Shaw Blvd., St. Louis, MO 63110, USA

**Keywords:** Amaryllidaceae, Amaryllidaceae alkaloids, *Hippeastrum*, GC-MS, alkaloid profiling, acetylcholinesterase, butyrylcholinesterase, Alzheimer’s disease

## Abstract

Amaryllidaceae alkaloids from the Amaryllidoideae subfamily exhibit broad pharmacological activities, including neuroprotection and anticancer effects. Galanthamine is a key compound for Alzheimer’s therapy. The *Hippeastrum* genus, particularly in Bolivia, offers significant potential for novel drug discovery, emphasising the need for conservation and further phytochemical research. Twenty-seven samples from Bolivian *Hippeastrum* species were investigated in terms of their alkaloid profile and anticholinesterase activity. The phytochemical analysis of Bolivian *Hippeastrum* species via GC-MS identified 48 Amaryllidaceae alkaloids, displaying diverse structural groups with potential pharmacological significance. Lycorine- and Homolycorine-type alkaloids were predominant, particularly in *H. chionedyanthum* and *H. haywardii*, with high concentrations of lycorine, a promising anticancer compound. The species *H. evansiarum* and *H. mollevillquense* contained notable quantities of Galanthamine type alkaloids, relevant for Alzheimer’s treatment. This study also highlights variability in acetylcholinesterase and butyrylcholinesterase inhibitory activities, with *H. lara-ricoi* and *H. haywardii* demonstrating strong inhibition. These findings suggest that *Hippeastrum* species are a valuable source of bioactive compounds, warranting further research into their therapeutic applications.

## 1. Introduction

Research into bioactive compounds extracted from natural sources is experiencing significant progress in the scientific field, especially in medicine for the development of new drugs. Alkaloids are recognised for their structural diversity and potent biological activities. Among the metabolites synthesised by plants, alkaloids have provided important biological benefits [1].

The subfamily Amaryllidoideae stands out for its richness in bioactive compounds, including Amaryllidaceae alkaloids (AAs), an exclusive alkaloid group only present in this plant subfamily. AAs stand out for presenting complex chemical structures and significant biological activities, such as antiparasitic, antiproliferative, antifungal, cytotoxic, psychopharmacological, and acetylcholinesterase inhibition activities [2,3,4]. The Amaryllidoideae subfamily is one of the 20 most relevant alkaloid-containing plant groups, comprising more than 800 perennial bulbous species classified into 59 genera [4,5].

AAs are derived from the aromatic amino acids phenylalanine and tyrosine and are synthesised within the norbelladine pathway [6]. Norbelladine is methylated to form 4′-*O*-*methylnorbelladine*, which serves as a key intermediate for multiple biosynthetic pathways. The phenolic oxidative coupling of this structure results in the formation of different AA groups [2,7,8]. In a recent review, AAs were classified into 42 types of skeletons, including protoalkaloids (present in numerous species of Amaryllidaceae), *Sceletium* alkaloids, and miscellaneous. More than 600 different AA structures are available in the Amaryllidoideae subfamily [4].

Galanthamine is the most recognised AA, and it has aroused great interest in the scientific community due to its unique pharmacological properties. It is used and commercialised for the palliative treatment of mild to moderate Alzheimer’s disease [9,10,11,12]. Its mechanism of action lies in the competitive and reversible inhibition of acetylcholinesterase (AChE), as well as being an allosteric ligand to nicotinic acetylcholine receptors (nAChRs) [13,14]. This enzyme is crucial in the degradation of acetylcholine, an essential neurotransmitter for memory and cognition. Cholinesterase therapy can improve brain function in neurodegenerative diseases, such as Alzheimer’s [8]. Beyond Alzheimer’s, preclinical studies have shown galanthamine’s potential in treating other neurodegenerative diseases, such as Parkinson’s [15]. Its anticancer activity is also promising, with studies linking it to tumour growth inhibition in ovarian cancer, colorectal adenocarcinoma, prostate cancer, pancreatic cancer, lung cancer, astrocytoma, glioma, human myeloid leukaemia, and hepatocellular carcinoma [16,17,18,19,20,21,22,23].

Galanthamine was initially isolated from *Galanthus woronowii* in the 1950s. Currently, it is mainly obtained from various species of Amaryllidaceae such as *Lycoris* sp. and ornamental *Narcissus* cultivars. Approved by the FDA in 2001, it is marketed under the names Reminyl^®^ in Europe and Razadyne^®^ in the United States, and also, the generic drug was commercialised following patent expiration [2].

One of the most promising genera within the Amaryllidaceae family is the genus *Hippeastrum*, known for its rich chemical diversity and traditional use in folk medicine to treat a variety of ailments [24,25]. Recent research has demonstrated the presence of a wide range of alkaloids exhibiting various biological activities, including antiparasitic, antioxidant, and neuroprotective functions [24,26,27,28,29,30]. Additionally, antimicrobial, anti-inflammatory, antioxidant, antitumoral, and neuroprotective properties have been reported for these genera [24,27,31,32,33,34]. These findings support the hypothesis that the bioactive compounds present in *Hippeastrum* could represent a valuable source of new therapeutic agents to address both infectious and neurodegenerative diseases.

The genus *Hippeastrum*, popularly known as amaryllis, stands out within the Amaryllidaceae family for its exceptional diversity of species. This wide variety of species has motivated the performance of numerous phytochemical studies, revealing the complex and fascinating richness of alkaloids that these plants possess, with significant bioactive properties. The chemistry and biological activity of *Hippeastrum* genus alkaloids were reviewed in [32] in 2012 and updated in [35] in 2021. In recent years the research on *Hippeastrum* alkaloids has attracted attention, and about 90 alkaloids and 24 unidentified structures have been described between 2012 and 2021, with Lycorine, Haemanthamine, Homolycorine, and Galanthamine types standing out as the most prevalent [35]. Additionally, Tallini’s study underscores the considerable biological potential of these alkaloids, particularly as cytotoxic agents, AChE inhibitors, and antiprotozoal agents.

Bolivia (South America) is home to 35 species of *Hippeastrum*, 24 of which are endemic to the country [36,37]. This fact positions Bolivia as one of the most important centres of diversity for this genus, alongside Peru, Brazil, and Argentina [38]. Bolivian *Hippeastrum* species are distributed across various ecosystems, from tropical rainforests to mountainous areas, adapting to altitudes ranging from 170 to 3000 m above sea level. This diversity is reflected in the wide range of flower shapes, colours, and sizes, with flowers whose colouration can switch from pure white to vibrant shades of red, yellow, orange, pink, and purple. The flowers are large, often measuring up to 20 cm in diameter, and trumpet shaped. The leaves are long and lanceolate, and the underground organs are bulbs. These plants reproduce by producing trilocular capsules containing numerous black seeds [36,37,39].

Bolivian *Hippeastrum* species represent a valuable treasure of biodiversity with substantial scientific and medicinal potential. Their study and conservation are essential for developing new medical treatments and for preserving this natural resource for future generations. Protected areas, such as Madidi National Park, Toro Toro National Park, Carrasco National Park, Cotapata National Park, and the Apolobamba Integrated Management Area, are crucial habitats for *Hippeastrum* species [40].

The aim of this study was to obtain the alkaloid profiles of 27 samples of *Hippeastrum* species collected in Bolivia, as well as to evaluate the potential of the samples in Alzheimer’s disease treatment.

## 2. Materials and Methods

### 2.1. Plant Material

Twenty-seven samples corresponding to 23 species of the genus *Hippeastrum*, collected in various geographical areas of Bolivia between December 2022 and February 2023, were analysed.

Healthy and representative specimens were selected, recording detailed data on the geographical location, habitat, and morphological characteristics of each plant. The samples were obtained from areas of high diversity of Bolivian flora. The samples include *H. cardenasii* (H1), *H. chionedyanthum* (H2), *H. cybister* (H3), *H. cybister* (H4), *H. escobaruriae* (H5), *H. evansiarum* (H6), *H. evansiarum* (H7), *H. fragrantissimum* (H8), *H. haywardii* (H9), *H. haywardii* (H10), *H. incachacanum* (H11), *H. lapacense* (H12), *H. lara-ricoi* (H13), *H. leopoldii* (H14), *H. mollevillquense* (H15), *H. nelsonii* (H16), *H. paquichanum* (H17), *H. pardinum* (H18), *H. parodii* (H19), *H. psittacinum* (H20), *H. puniceum* (H21), *H. puniceum* (H22), *H.* sp. (H23)*, H. umabisanum* (H24), *H. vittatum* (H25), *H. warszewiczianum* (H26), and *H. yungacense* (H27). Photographic documentation of the species studied in this work is shown in Figure 1.

The samples were collected by prioritising areas with a high diversity of flora. *Hippeastrum* species are distributed in various regions and specific habitats in Bolivia. For example, *H. cardenasii* (H1) is found in the Madidi National Park in La Paz, characterised by its humid climate and well-drained soils. In contrast, *H. chionedyanthum* (H2) inhabits the Amboró National Park in Santa Cruz, where clay soil and moderate humidity conditions predominate. *Hippeastrum cybister* is divided into two variants: one of them is observed on the Eagle Trail in La Paz and is adapted to sandy soils and high humidity (H3). The other variant thrives in Tiquipaya, Cochabamba, in soils rich in organic matter (H4). *H. evansiarum* (H6) is found in Andrés Ibáñez, Santa Cruz, in areas with well-drained and moderately moist soils, while *Hippeastrum evansiarum* (H7) lives in La Angostura, Santa Cruz, in soils with sandy soils and low to moderate humidity. The location of each *Hippeastrum* sample is shown in Figure 2, and the geographic and climatic characteristics of the collection sites are detailed in Table 1.

Taxonomic identification was carried out with the expertise of Raúl Rico Lara, regarded as the leading expert on the *Hippeastrum* genus in Bolivia, Margoth Atahuachi, Alfredo Fuentes, and Carla Maldonado. Additionally, Moisés Mendoza (Herbario del Oriente Boliviano (USZ) and Museo de Historia Natural Noel Kempff Mercado) contributed to the taxonomic verification process of two specimens.

Each specimen was thoroughly analysed, documenting detailed collection information, morphological characteristics, and environmental conditions. The collection and taxonomic characterisation of the plant material were carried out as part of the project “Bioprospecting, chemical characterisation and biological evaluation of *Amaryllidaceae* plants endemic to different regions of Bolivia”, developed in collaboration with the Institute of Ecology of the Universidad Mayor de San Andrés (UMSA).

The detailed identification of the specimens, as well as the Collection Code, the origin of the samples, and the personnel responsible for collection and identification, is detailed in Table 2.

### 2.2. Alkaloid Extraction

Purified alkaloid extracts were prepared from plant samples following previously procedure described in the literature [41]. Initially, the samples were cut and dried at 40 °C and then pulverised in a stainless-steel rotary knife mill. The resulting powder was macerated in HPLC-grade methanol (50 mL) at 25 °C for three days, adding 50 mL of fresh methanol daily for continuous extraction. An ultrasonic bath (20 min, 4 times a day) was applied to improve alkaloid extraction. After each treatment, the solution was filtered, and the solvent was evaporated in a vacuum to obtain crude extracts.

The crude extracts were collected separately after each day of maceration and ultrasonic treatment. Subsequently, they were acidified with 2% sulfuric acid (*v*/*v*) (50 mL) to adjust the pH to 2 and subjected to treatment with ethyl acetate (3 × 50 mL) to eliminate neutral compounds. The aqueous phase pH was then adjusted to 9–10 with 25% ammonium hydroxide (*v*/*v*), followed by extraction with ethyl acetate (3 × 50 mL). After evaporating the solvent, the dry alkaloid extract was obtained and used for chemical characterisation and the evaluation of its pharmacological activity. The extraction yield is shown in Table 3.

### 2.3. GC-MS Analysis

Two mg of each alkaloid extract was dissolved in 1 mL of methanol containing 25 μg·mL^−1^ codeine as an internal standard. One μL of the mixture was injected into a gas chromatograph (Agilent Technologies 6890N, Santa Clara, CA, USA) coupled to a mass spectrometer (Agilent Technologies 5975, Santa Clara, CA, USA). The system used electronic impact ionisation (EI) at 70 eV and a Series 7683B automatic injector (Agilent Technologies, Santa Clara, CA, USA). A Tecknokroma TR-45232 Sapiens-X5MS column (30 m × 0.25 mm, film thickness 0.25 μm) and splitless-mode injection were used. The temperature program included an initial 12 min increment at 100 °C, followed by an increase to 180 °C at a rate of 15 °C/min, maintenance for 1 min at 180 °C, an increase to 300 °C at 5 °C/min, and a 10 min period at 300 °C. The injector and detector temperatures were 250 °C and 280 °C, respectively, with a carrier gas flow (He) of 1 mL/min.

### 2.4. Identification of Alkaloids

The alkaloids were identified by comparing their mass spectra and their Kovats Retention Indices (RIs) with those of the authentic standards of the Amaryllidaceae alkaloid library of the Natural Products Research Group of the Faculty of Pharmacy of the University of Barcelona, which includes more than 300 alkaloids of Amaryllidaceae, identified using spectroscopic techniques such as NMR, CD, MS, and IR. The mass spectra were deconvoluted using Automatic Mass spectral Deconvolution and Identification System (AMDIS 2.64) software from NIST (Gaithersburg, MD, USA).

### 2.5. Quantification of Alkaloids

For the relative quantification of alkaloids, a galanthamine calibration curve (at concentrations of 10, 20, 40, 60, 80, and 100 μg·mL^−1^) was used, with codeine (25 μg·mL^−1^) as the internal standard. The area of the deconvoluted peaks was determined for relative quantification using Excel 2016 software. This is not an absolute quantification but serves as a comparison of the relative quantity of alkaloids between different samples that have been extracted, purified, and analysed using the same methodology [41].

### 2.6. AChE and Butyrylcholinesterase (BuChE) Inhibition Assays

The inhibitory potential of *Hippeastrum* extracts on AChE and BuChE enzymes was evaluated by a colourimetric assay based on the formation of thiobenzoate anion (yellow) after the reaction of thiocholine and 5,5′-dithio-*bis*-(2-nitrobenzoic acid) (DTNB). The method used was an adaptation of the protocol described in [42], with adjustments in accordance with the guidelines previously established in [43].

Stock solutions of AChE enzymes from *Electrophorus electricus* and BuChE from equine serum (518 U/mL) were prepared and stored at −20 °C. The reagents used included DTNB, *S*-butyrylthiocholine iodide (BTCI), and acetylthiocholine iodide (ATCI) and were supplied by Merck (Darmstadt, Germany).

The trial began with the mixture of 50 μL of AChE or BuChE (both enzymes at 6.24 U in phosphate buffer (8 mM K_2_HPO_4_, 2.3 mM NaH_2_PO_4_, and 0.15 M NaCl, pH 7.5)) and 50 μL of the alkaloid extract dissolved in the same buffer. The plates were incubated for 30 min at 25 °C. Subsequently, 100 μL of the substrate solution (0.1 M Na_2_HPO_4_, 0.2 M DTNB, and 0.6 mM ATCI or 0.24 mM BTCI in Millipore water, adjusted to pH 7.5) was added. Absorbance was read at 405 nm after 10 min using a Labsystem microplate reader (Helsinki, Finland). Enzyme activity was calculated as a percentage relative to a control containing only the buffer, with no inhibitor. Galanthamine was used as a positive control in this study, with concentrations of 0.1, 0.2, 0.3, 0.4, 0.5, 1.0, and 2.0 μg·mL^−1^ for AChE inhibition and 1, 4, 6, 8, 10, 12, and 15 μg·mL^−1^ for BuChE inhibition. The samples were screened using specific concentrations for each species to determine their IC_50_ values, ranging from 0.1 to 250 μg·mL^−1^ for AChE and from 5 to 250 μg·mL^−1^ for BuChE. The concentration ranges were established based on the results of a preliminary dose–response analysis. Three replicates were performed. The data were analysed using PRISM 10 software (Boston, MA, USA) and are reported as the mean ± standard deviation (SD). One-way ANOVA was performed, followed by Dunnett’s multiple-comparisons test, to evaluate differences in response to galantamine in relation to both AChE and BuChE.

## 3. Results

### 3.1. Identification and Quantification of Bioactive Compounds

The phytochemical analysis conducted on various *Hippeastrum* species revealed a remarkable diversity of alkaloids, classified into several structural groups. The identification of 48 known Amaryllidaceae alkaloids suggests complex distribution patterns that may relate to specific defence mechanisms or ecological adaptations to growth conditions. The alkaloids identified in Bolivian *Hippeastrum* species are presented in Table 4. The entire quantification results for each sample are detailed in the Appendix A and briefly described below.

#### 3.1.1. Lycorine Type Alkaloids

Lycorine type alkaloids were predominantly found in some species of *Hippeastrum*. *H. chionedyanthum* (**H2**) stands out for presenting an exceptionally high quantity of Lycorine type compounds (80.8 μg Gal/100 mg DW), mostly owing to the high amount of galanthine (39.1 μg Gal/100 mg DW) and tortuosine (20.6 μg Gal/100 mg DW).

*H. yungacense* (**H27**), *H. haywardii* (**H9**), and *H. mollevillquense* (**H15**) also present a notable quantity of Lycorine type compounds, with a total of 59.2, 49.4, and 33.1 μg Gal/100 mg DW, respectively. **H27** and **H9** present mostly lycorine, with significant values of this alkaloid (39.8 and 25.2 μg Gal/100 mg DW), respectively. Lycorine is attracting attention and standing out as a promising compound in anticancer therapy owing to its potent action in vitro and in vivo against other drug-resistant cancer cells [44,45,46,47]. The high concentration of Lycorine type compounds and, specifically, of lycorine in these plant species suggests that they could be an important source of these compounds for pharmacological studies. **H15** also presents a considerable amount of dihydrolycorine (16.5 μg Gal/100 mg DW).

In general, Lycorine type alkaloids appear to be distributed variably among species, with a tendency to concentrate in the species that exhibit greater adaptability to more stressful environments, such as **H2** and **H9**. This variability in the concentration of lycorine and its derivatives could represent a defence mechanism against pests or environmental stresses.

#### 3.1.2. Homolycorine Type Alkaloids

Homolycorine type alkaloids are the predominant class of alkaloids in Bolivian *Hippeastrum* species. Among these, homolycorine is the most abundant compound, particularly in *H. fragantissimum* (**H8**) with 36.4 μg Gal/100 mg DW. This species is also the species with a higher amountof Homolycorine type compounds (60.4 μg Gal/100 mg DW), with the additional presence of a significant quantity of nerinine (24.0 μg Gal/100 mg DW).

*Hippeastrum escobaruriae* (**H5**) showed a high concentration of candimine (15.01 µg Gal/100 mg DW), an alkaloid known for its antimicrobial activity [48].

Both samples of *H. cybister* (**H4** and **H3**) also show a notable amount of Homolycorine type compounds, with 37.6 and 28.9 µg Gal/100 mg DW, respectively.

Homolycorine, 8-*O*-demethylhomolycorine, hippeastrine, and candimine have demonstrated activities such as cytotoxic, antiretroviral, hypotensive, and antifungal, but the bioactivity of most Homolycorine type compounds still remains unknown [32]. However, this type of alkaloid seems to play an important role in the defence of *Hippeastrum* plants against pathogens [49].

#### 3.1.3. Galanthindol Type Alkaloids

The only member of this group of alkaloids detected was galanthindol, and it was especially abundant in *H. vittatum* (**H25**), with 16 µg Gal/100 mg DW. This alkaloid was isolated from *Galanthus plicatus* ssp. *byzantinus* (Amaryllidaceae), a plant native to northwestern Turkey [50]. No studies have been found documenting specific biological properties of galanthindol, and more research is needed to understand its potential and characteristics.

#### 3.1.4. Haemanthamine/Crinine Type Alkaloids

The presence of Hamanthamine/Crinine type alkaloids is irregularly distributed among different Bolivian *Hippeastrum* species. While there are several species without any alkaloids of this type, some others present a significant quantity of them. *H. warszewiczianum* (**H26**) stands out for showing the highest amount of this type of compounds (45.9 µg Gal/100 mg DW), mostly due to the presence of an important content of haemanthamine (22.9 µg Gal/100 mg DW). Although not in such quantity, it also shows the presence of 8-*O*-demethylmaritidine, vittatine/crinine, and 3-crinane-3-one (8.3, 7.4, and 7.3 µg Gal/100 mg DW, respectively). *H. haywardii* (**H9**) also presents Haemanthamine/Crinine type alkaloids (26.0 µg Gal/100 mg DW), with demethylmaritidine (14.0 µg Gal/100 mg DW) and vittatine/crinine (12.0 µg Gal/100 mg DW). Finally, another species with remarkable content of this type of compound is *H. lara-ricoi* (**H13**) (23.6 µg Gal/100 mg DW), with *O*-methylmaritidine (15.5 µg Gal/100 mg DW) and maritidine (8.1 µg Gal/100 mg DW). Haemanthamine has demonstrated to induce apoptosis in tumour cells and antimalarial activity [51,52].

#### 3.1.5. Narciclasine Type Alkaloids

Trisphaeridine is the only alkaloid of this type found in *Hippeastrum* species. It is detected in *H. haywardii* (**H10**), *H. paquichanum* (**H17**), and *H. escobaruriae* (**H5**) at 16.1, 15.9 and 14.8 µg Gal/100 mg DW, respectively. Narciclasine type alkaloid compounds have shown potential anticancer properties with promising therapeutic applications [53]. Furthermore, trisphaeridine has recently be found to exhibit inhibitory activities against Tobacco mosaic virus [54].

#### 3.1.6. Pretazettine Type Alkaloids

Some Bolivian *Hippeastrum* species are especially rich in Pretazzetine type alkaloids. *H. paquichanum* (**H17**) is the species with the highest content of this type of alkaloid (54.5 µg Gal/100 mg), owing to the presence of *O*-methyltazettine (38.6 µg Gal/100 mg DW) and tazettine (15.9 µg Gal/100 mg DW). *H. vittatum* (**H25**) also shows a notable amount of this type of alkaloid (35.7 µg Gal/100 mg DW), with tazettine, *O*-methyltazettine, and 3-*epi*-macronine (12.5, 7.8, and 7.7 µg Gal/100 mg DW, respectively). The *H. escobaruriae* (**H5**) content of Pretazettine type alkaloids is 30.7 µg Gal/100 mg DW, owing to the presence of tazettine and 3-*epi*-macronine (15.9 and 14.8 µg Gal/100 mg DW, respectively). These compounds have potential applications in the development of anticancer drugs, as they have shown cytotoxicity against different types of neoplastic cell lines [2].

#### 3.1.7. Montanine Type Alkaloids

Two *Hippeastrum* species are especially rich in Montanine type compounds: *H. umabisanum* (**H24**) and *H. evansiarum* (**H7**) (28.0 and 27.9 µg Gal/100 mg DW, respectively). The most abundant Montanine type compound is 2-*O*-methylpancracine (19.1 µg Gal/100 mg DW in H24). Pancratinine C appears also in a high quantity in *H. mollevillquense* (**H15**), with 16.8 µg Gal/100 mg DW. Montanine type alkaloids represent an interesting type of compounds owing to their remarkable broad spectrum of pharmacological activities, such as anxiolytic, antidepressant, anticonvulsant, inhibition of AChE, antibacterial, and antiparasitic [32,55].

#### 3.1.8. Galanthamine Type Alkaloids

Galanthamine type alkaloids show heterogeneous distribution across species. In *H. evansiarum* (**H7**), the total content reaches 38.59 µg Gal/100 mg DW, with lycoramine as the main compound (23.8 µg Gal/100 mg DW) followed by norlycoramine (14.8 µg Gal/100 mg DW). *H. mollevillquense* (**H15**) also exhibits a significant content of Galanthamine type alkaloids (36.5 µg Gal/100 mg DW), owing to the presence of galanthamine and narwedine (20.0 and 16.5 µg Gal/100 mg DW, respectively). Finally, *H. yungacense* (**H27**) shows, as well, a good quantity of Galanthamine type compounds (33.0 µg Gal/100 mg DW), owing to the presence of galanthamine (14.5 µg Gal/100 mg DW), narwedine (9.2 µg Gal/100 mg DW), and lycoramine (9.3 µg Gal/100 mg DW). Galanthamine is a well-known Amaryllidaceae alkaloid commercialised for the palliative treatment of the mild and moderate stages of Alzheimer’s disease [10].

#### 3.1.9. Ismine Type Alkaloids

Ismine is the only Ismine type alkaloid detected in Bolivian *Hyppeastrum* species, and it is distributed, principally, in *H. escobaruriare* (**H5**) (15 µg Gal/100 mg DW), *H. vittatum* (**H25**) (12.9 µg Gal/100 mg DW), *H. evansiarum* (**H7**) (11.8 µg Gal/100 mg DW), and *H. haywardii* (**H9**) (11.1 µg Gal/100 mg DW). Ismine has reported hypotensive activity in rats and cytotoxicity against Molt 4 lymphoid and LMTK fibroblastic cell lines [2].

### 3.2. AChE and BuChE Inhibitory Activities

The inhibitory activities of AChE and BuChE were evaluated in alkaloid extracts from various *Hippeastrum* species. The results revealed considerable variability in the IC_50_ values, reflecting significant differences in the inhibitory potency among the samples. The results are reported in Table 5.

*H. lara-ricoi* (**H13**) exhibited the strongest inhibition of AChE, with an IC_50_ of 2.32 μg·mL^−1^, and also a good inhibition of BuChE, with an IC_50_ of 53.15 μg·mL^−1^. *H. haywardii* (**H9**) demonstrated high inhibitory activity of both enzymes, with an IC_50_ of 3.54 μg·mL^−1^ for AChE and 23.26 μg·mL^−1^ for BuChE.

*H. chionedyanthum* (**H2**) and *H. yungacense* (**H27**) also demonstrated significant anti-AChE activity, with IC_50_ values of 6.03 and 6.42 μg·mL^−1^, respectively. *H. yungacence* also exhibited strong BuChE inhibitory activity, with an IC_50_ of 35.97 μg·mL^−1^.

Several other species exhibited moderate inhibition profiles with both enzymes. *H. cardenasii* (**H1**) showed an IC_50_ of 49.67 μg·mL^−1^ for AChE and 149.13 μg·mL^−1^ for BuChE. For *H. nelsonii* (**H16**), the IC_50_ values were 51.66 μg·mL^−1^ for AChE and 116.02 μg·mL^−1^ for BuChE. *H. fragrantissimum* **(H8)** recorded an IC_50_ of 11.85 μg·mL^−1^ for AChE and 103.62 μg·mL^−1^ for BuChE, while *H. vittatum* **(H25)** had an IC_50_ of 36.68 μg·mL^−1^ for AChE and 169.65 μg·mL^−1^ for BuChE. *H.* sp. (**H23**) and *H. parodii* (**H19**) also exhibited inhibitory activity for both enzymes, with AChE IC_50_ values of 33.58 and 35.13 μg·mL^−1^ and BuChE IC_50_ values of 111.70 and 119.23 μg·mL^−1^, respectively.

*H. mollevillquense* **(H15)** and *H. warszewiczianum* **(H26),** although not as notable as previous samples, also exhibited activity inhibiting AChE, with IC_50_ values of 36.52 and 47.43 μg·mL^−1^, respectively.

## 4. Discussion

The phytochemical analysis of various *Hippeastrum* species revealed a diverse and complex array of bioactive alkaloids, each contributing to the ecological and pharmacological significance of these plants. The identification of 45 known Amaryllidaceae alkaloids highlights the intricate distribution patterns of these compounds across species, which may be linked to specific defence mechanisms or ecological adaptations to their environments. These findings suggest that Bolivian *Hippeastrum* species could be promising candidates for the development of new therapeutic agents targeting a variety of diseases.

Homolycorine-, Pretazettine-, and Lycorine-type alkaloids were the most abundant alkaloid type in Bolivian *Hippeastrum* species (Figure 3). Homolycorine type compounds were mostly represented by candimine and nerinine, with *Hippeastrum fragrantissimum* (**H8**) showing a particularly high concentration of nerinine. The antimicrobial and cytotoxic properties of Homolycorine type alkaloids further emphasise the therapeutic potential of *Hippeastrum* species in the treatment of infections and other diseases. Pretazettine type alkaloids were specially represented by *O*-methyltazettine, tazettine, and 3-epimacronine, being particularly found in *Hippeastrum escobaruriae* (**H5**) and *H. paquichanum* (**H17**). Given the cytotoxic activity reported by these kinds of compounds, these species could be in consideration for future applications in the development of new anticancer drugs.

Lycorine type alkaloids were particularly abundant in *Hippeastrum chionedyanthum* (**H2**) and *H. haywardii* (**H9**). The high concentrations of lycorine and its derivatives, such as galanthine and tortuosine, underscore their potential as a source of bioactive compounds with potent anticancer properties. Considering lycorine’s promising activity against drug-resistant cancer cells, these species may be valuable for future pharmacological studies focused on cancer therapy. The variable distribution of Lycorine type alkaloids among *Hippeastrum* species also suggests that these compounds may play a role in plant defence, potentially acting as deterrents against herbivores or pathogens in species exposed to more stressful environments, or be an adaptation to survive in more arid habitats.

Galanthamine type alkaloids were present in notable quantities in *Hippeastrum evansiarum* (**H7**) and *H. mollevillquense* (**H15**), particularly galanthamine.

Galanthamine, lycorine, and tazettine are the most commonly occurring Amaryllidaceae alkaloids in non-Bolivian *Hippeastrum* species. Galanthamine has been detected in *H. argentinum*, *H. aulicum*, *H. equestre*, *H. glaucescens*, *H. goianum*, *H. morelianum*, *H. papilio*, *H. rutilum*, *H. santacatarina*, and *H. solandriflorum* [32,35,56]. Tazettine has been identified in *H. aulicum*, *H. johnsonii*, *H. equestre*, *H. glaucescens*, *H. morelianum*, *H. santacatarina*, and *H. breviflorum* [32]. Lycorine has been reported in *H. argentinum*, *H. bifidum*, *H. rutilum*, *H. rachyandrum*, *H. aulicum*, *H. candidum*, *H. johnsonii*, *H. añañuca*, *H. bicolor*, *H. equestre*, *H. solandriflorum*, *H. glaucescens*, *H. striatum*, and *H. santacatarina* [32,56].

To the best of our knowledge, this study represents the first report of the following alkaloids in *Hippeastrum* species: 2-methoxy-8-*O*-demethylhomolycorine, 2-*O*-methylpancracine, 3-*O*-acetylgalanthamine, 3-*O*-demethyltazettine, cliviasine, crinane-3-one, demethylismine, dihydrolycorine, epigalanthamine, flexinine, hippeastidine, isotazettino, lycoraminone, maritidine, norlycoramine, *O*-methylmaritidine, *O*-methyltazettine, tortuosine, and undulatine.

The species *H. lara-ricoi* (**H13**), in particular, demonstrated the strongest AChE inhibition, suggesting that it could be a key species for further research in this area.

The variability in AChE and BuChE inhibitory activities across *Hippeastrum* species further reinforces the idea that these plants contain a wide range of bioactive compounds with diverse pharmacological effects. *H. haywardii* (**H9**) demonstrated high inhibitory activity against AChE. However, it does not contain Galanthamine type compounds, On the other hand, **H9** contains a high amountof Lycorine type compounds, especially lycorine. The same situation occurred with *H. lara-ricoi* (**H13**), demonstrating good inhibitory activity of both enzymes without Galanthamine type alkaloids. These findings suggest that other compounds different than Galanthamine type alkaloids may contribute to the inhibition of these enzymes in *Hyppeastrum* species. On the other hand, the exceptional AChE inhibition activity demonstrated by **H13** suggests that it could be a key species for further research in this area. *H. yungacense* (**H27**) also reports good results inhibiting both enzymes. In this case, such activity could be attributed to the notable quantity of Galanthamine type alkaloids, especially galanthamine.

Synergism is a factor that must also be considered in this type of analysis, as it significantly influences the therapeutic effectiveness of herbal medicine [57]. For instance, cherylline has been shown to exhibit synergistic effects with other AAs, resulting in enhanced inhibition of dengue virus replication [58]. Synergistic inhibition of acetylcholinesterase has been observed with combinations of various alkaloids from Chinese herbal medicines [59]. Similarly, synergistic antiparasitic activity of *Hippeastrum* alkaloids has been reported, particularly when montanine is combined with benznidazole, resulting in a potent effect against *Trypanosoma cruzi* [30].

Since some plants have shown promising in vitro assay results, they could be considered suitable candidates for future in vivo studies to assess their potential pharmacological properties and therapeutic applications.

Chemical variability in *Hippeastrum* alkaloids is influenced by factors such as soil composition, climate, and genetic traits [2,60]. Therefore, an in-depth biological evaluation of these compounds promises to open new perspectives for the scientific and industrial exploitation of this genus [35].

## 5. Conclusions

To the best of our knowledge, the present research represents the first description of pharmacological activity and alkaloid characterisation of Bolivian *Hippeastrum* species. Furthermore, it is also the first report of the phytochemical characterisation of all the species, except for *H. puniceum* (**H21**), *H. psittacinum* (**H20**), and *H. vittatum* (**H25**) [32].

The diversity of alkaloids found across *Hippeastrum* species not only highlights the chemical richness of this genus but also underscores its ecological and pharmacological importance. According to the compounds identified, Bolivian *Hippeastrum* species can be a valuable resource for the development of new therapeutic agents. The complex distribution of these alkaloids across species may also reflect their adaptive strategies in response to environmental pressures, further emphasising the intricate relationship between plant chemistry and ecological function.

## Figures and Tables

**Figure 1 life-15-00719-f001:**
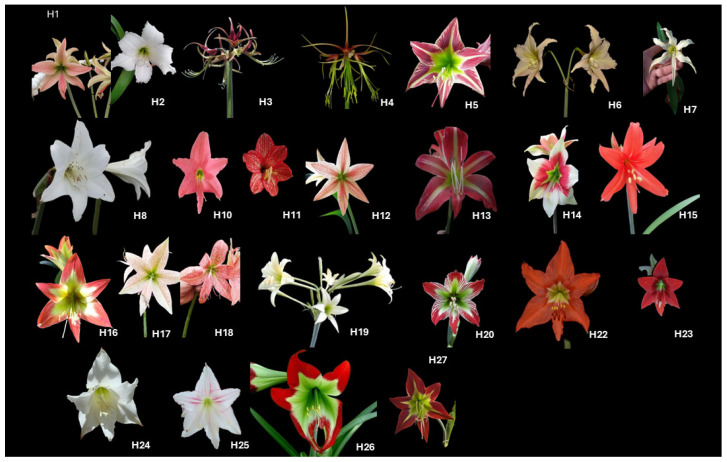
Photographic documentation of the flowers of *Hippeastrum* species from Bolivia: *H. cardenasii* (H1), *H. chionedyanthum* (H2), *H. cybister* (H3), *H. cybister* (H4), *H. escobaruriae* (H5), *H. evansiarum* (H6), *H. evansiarum* (H7), *H. fragrantissimum* (H8), *H. haywardii* (H9), *H. haywardii* (H10), *H. incachacanum* (H11), *H. lapacense* (H12), *H. lara-ricoi* (H13), *H. leopoldii* (H14), *H. mollevillquense* (H15), *H. nelsonii* (H16), *H. paquichanum* (H17), *H. pardinum* (H18), *H. parodii* (H19), *H. psittacinum* (H20), *H. puniceum* (H22), *H.* sp. (H23), *H. umabisanum* (H24), *H. vittatum* (H25), *H. warszewiczianum* (H26), and *H. yungacense* (H27). Source: Photographs were taken by the authors, collaborators, and collectors. H24 and H26 are adapted from [36].

**Figure 2 life-15-00719-f002:**
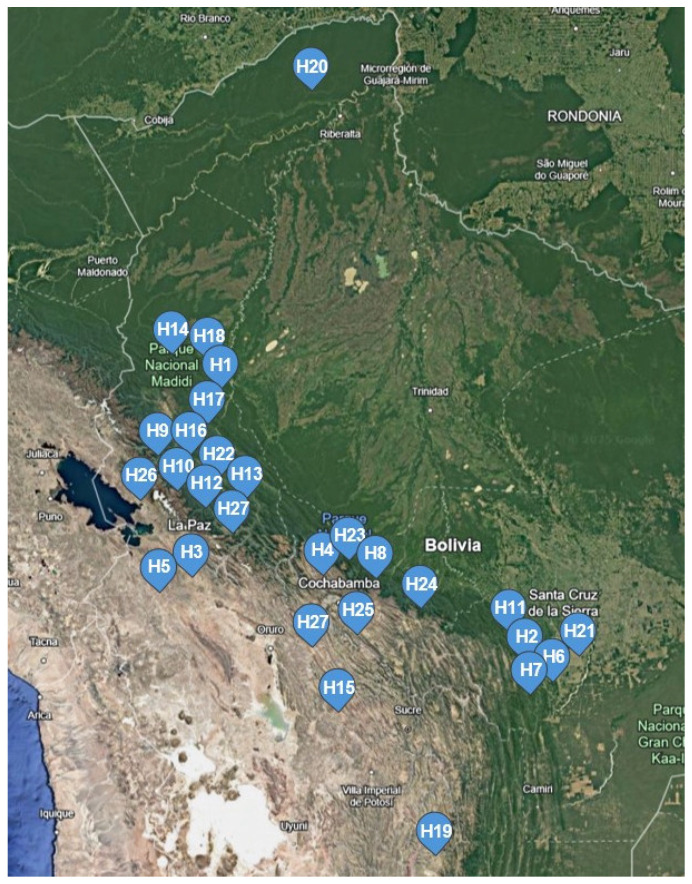
Geographic localisation of collection sites of different species of *Hippeastrum* in Bolivia, as referenced by Table 2 codes. Source: Google Earth.

**Figure 3 life-15-00719-f003:**
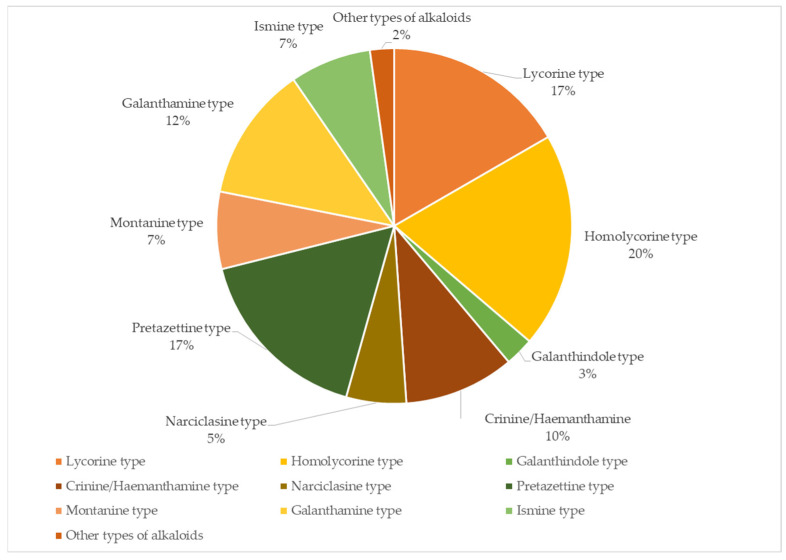
Alkaloid type characterisation in Bolivian *Hippeastrum* species. The values represent the percentage of each alkaloid type relative to the total alkaloid content, expressed in µg Gal/100 mg DW.

**Table 1 life-15-00719-t001:** Geographical and climatic characteristics of the collection sites.

Code	Altitude (m.a.s.l.)	Average Annual Precipitation (mm)	Soil Type	Average Annual Temperature (°C)
H1, H14, H17, H18	300–500	1800–2000	clayey	20–25
H2, H6, H21	200–600	1200–1400	sandy	22–28
H3	3600	800	rocky	15–10
H4, H8, H23	2500–2800	1300–1500	alluvial	18–22
H5, H12, H16	1800–2920	1500–3000	silty	15–20
H7	200	1200	sandy	23–30
H9	2200	1200	clayey	14–18
H10	2500	1500	sandy	17–22
H11	500	1100	rocky	20–25
H13	1800–2200	1500	alluvial	16–20
H15	2800	800	rocky	15–10
H19	2500	1200	rocky	15–20
H20	200	2000	clayey	24–30
H22	500	1600	silty	22–28
H24	2200	1300	alluvial	18–22
H25	2300	1100	clayey	19–23
H26	2800	1200	rocky	16–10
H27	2000	1000	alluvial	15–19

**Table 2 life-15-00719-t002:** Species of *Hippeastrum* collected in Bolivia and selected for the chemical and biological characterisation.

Code	Species	Collection Code	Botanical Validation	Herbarium	Geographical Location
**H1**	*Hippeastrum cardenasii*	ACE131	[RL], [MA]	BOLV	La Paz (P.N. Madidi), Bolivia
**H2**	*Hippeastrum chionedyanthum*	ACE124	[RL], [MA]	BOLV	Santa Cruz (Bermejo), Bolivia
**H3**	*Hippeastrum cybister*	MLRE01	[AF]	LPB	La Paz (Sendero del Águila), Bolivia
**H4**	*Hippeastrum cybister*	ACE121	[RL], [MA]	BOLV	Cochabamba (Tiquipaya), Bolivia
**H5**	*Hippeastrum escobaruriae*	ACE111	[RL], [MA]	BOLV	La Paz (Coroico), Bolivia
**H6**	*Hippeastrum evansiarum*	MM3849	[MM]	USZ	Santa Cruz (P.N. Madidi), Bolivia
**H7**	*Hippeastrum evansiarum*	ACE118	[RL], [MA]	BOLV	Santa Cruz (La Angostura), Bolivia
**H8**	*Hippeastrum fragrantissimum*	ACE113	[RL], [MA]	BOLV	Cochabamba (Tablasmonte), Bolivia
**H9**	*Hippeastrum haywardii*	MM5811	[AF], [CM]	LPB	La Paz (Larecaja), Bolivia
**H10**	*Hippeastrum haywardii*	ACE 140	[RL], [MA]	BOLV	La Paz (Guanay), Bolivia
**H11**	*Hippeastrum incachacanum*	ACE 125	[RL], [MA]	BOLV	Santa Cruz (Piedramesa), Bolivia
**H12**	*Hippeastrum lapacense*	ACE 128	[RL], [MA]	BOLV	La Paz (Coroico), Bolivia
**H13**	*Hippeastrum lara-ricoi*	ACE132	[RL], [MA]	BOLV	La Paz (Sud Yungas), Bolivia
**H14**	*Hippeastrum leopoldii*	ACE139	[RL], [MA]	BOLV	La Paz (P.N. Madidi), Bolivia
**H15**	*Hippeastrum mollevillquense*	ACE129	[RL], [MA]	BOLV	Potosí (Mollevillque), Bolivia
**H16**	*Hippeastrum nelsonii*	ACE109	[RL], [MA]	BOLV	La Paz (Guanay), Bolivia
**H17**	*Hippeastrum paquichanum*	ACE138	[RL], [MA]	BOLV	La Paz (P.N. Madidi), Bolivia
**H18**	*Hippeastrum pardinum*	ACE137	[RL], [MA]	BOLV	La Paz (P.N. Madidi), Bolivia
**H19**	*Hippeastrum parodii*	ACE116	[RL], [MA]	BOLV	Chuquisaca (Inca Huasi), Bolivia
**H20**	*Hippeastrum psittacinum*	ACE133	[RL], [MA]	BOLV	Pando, Bolivia
**H21**	*Hippeastrum puniceum*	AF25733	[AF], [CM]	LPB	Santa Cruz (Andrés Ibáñez), Bolivia
**H22**	*Hippeastrum puniceum*	MM	[MM]	USZ	Santa Cruz (Jardín Botánico), Bolivia
**H23**	*Hippeastrum* sp.	ACE 105	[RL], [MA]	BOLV	Cochabamba (Tablasmonte), Bolivia
**H24**	*Hippeastrum umabisanum*	ACE152	[RL], [MA]	BOLV	Chuquisaca, Bolivia
**H25**	*Hippeastrum vittatum* var. *tweedianum*	ACE107	[RL], [MA]	BOLV	Cochabamba (Naranjitos), Bolivia
**H26**	*Hippeastrum warszewiczianum*	ACE153	[RL], [MA]	BOLV	La Paz (Sorata), Bolivia
**H27**	*Hippeastrum yungacense*	AF25732	[AF], [CM]	LPB	La Paz (Tamanpaya), Bolivia

The information in parentheses indicates the region of collection. P.N. = National Park. Collectors: ACE = Alex Cespedes Escalera, AF = Alfredo Fuentes, MM = Moisés Mendoza, and MLRE = Ma. Lenny Rodriguez Escobar. Botanical Validators: [RL] = Raúl F. Lara Rico, [MA] = Margoth Atahuachi Burgos, [AF] = Alfredo Fuentes, [CM] = Carla Maldonado, and [MM] = Moisés Mendoza. BOLV, Herbario Nacional Forestal “Martin Cardenas”; LPB, Herbario Nacional de Bolivia; USZ, Herbario del Oriente Boliviano.

**Table 3 life-15-00719-t003:** Extraction yield of different Bolivian *Hippeastrum* samples’ alkaloid extracts, expressed as mg/g DW.

Code	Extraction Yield (mg/g DW)
**H1**	2.0
**H2**	6.7
**H3**	6.3
**H4**	3.0
**H5**	11.7
**H6**	1.9
**H7**	5.0
**H8**	24.0
**H9**	15.0
**H10**	20.8
**H11**	1.0
**H12**	9.0
**H13**	9.8
**H14**	11.8
**H15**	19.0
**H16**	5.0
**H17**	14.0
**H18**	0.9
**H19**	4.0
**H20**	4.0
**H21**	4.1
**H22**	1.0
**H23**	6.0
**H24**	5.1
**H25**	10.0
**H26**	10.1
**H27**	10.0

**Table 4 life-15-00719-t004:** Summary of alkaloids identified in Bolivian *Hippeastrum* species via GC-MS, showing their Retention Index (RI), Retention Time (RT), and Base Ion level in *m*/*z* [M^+^].

Alkaloids	RI	RT (min)	[M^+^]
**Lycorine type**			
Anhydrolycorine	2818.0	24.6375	250
Assoanine	2909.0	25.7095	266
Dihydrolycorine	2745.0	23.7785	288
Galanthine	3022.4	27.0459	242
Lycorine	3138.3	28.4106	226
Tortuosine	3212.8	29.2889	296
11,12-Dehydroanhydrolycorine	2948.8	26.1784	248
**Homolycorine type**			
Candimine	3305.2	30.3770	125
Cliviasine	3181.0	28.9140	96
Hippeastrine	3269.9	29.9614	125
Homolycorine	2624.8	22.3619	109
Nerinine	3177.9	28.8780	109
*8*-*O*-Demethylhomolycorine	2739.7	27.0306	109
2-Hydroxyhomolycorine	3329.0	30.6576	109
2-Methoxy-8-*O*-demethylhomolycorine	3021.1	27.0308	109
**Galanthindole type**			
Galanthindole	2553.9	24.1963	281
**Crinine/Haemanthamine type**			
Maritidine	2566.1	21.6712	287
*O*-Methylmaritidine	2761.6	23.9741	301
Vittatine/crinine	2769.7	24.0685	271
*8*-*O*-Demethylmaritidine	2810.6	24.5511	273
Crinane-3-one	2852.4	27.5250	271/181
Crinamine	2945.0	26.1345	272
Hippeastidine	2947.4	26.1626	319
Haemanthamine	3008.2	26.8789	272
Flexinine	3042.0	27.2765	258
Undulatine	3123.8	28.2400	331
**Narciclasine type**			
Trisphaeridine	2346.1	20.9836	223
**Pretazettine type**			
Isotazettino	3077.1	27.6901	247
3-*O*-Demethyltazettine	3049.9	27.3696	247
3-epi-Macronine	3186.6	28.9797	245
Tazettine	2991.6	26.6831	247
*O*-Methyltazettine	2935.3	26.0200	261
5-Methoxyhomolycorine	3346.8	30.8677	139
**Montanine type**			
Pancracine	3027.5	27.1062	287
Pancratinine C	2893.0	25.5216	287
2-*O*-Methylpancracine	2967.6	26.4007	301
**Galanthamine type**			
Sanguinine	2666.4	22.8524	273
Galanthamine	2406.0	22.5833	286
Narwedine	2746.7	23.7978	284
3-*O*-Acetylgalanthamine	2824.7	24.7172	270
Lycoramine	2699.5	23.2424	288
Lycoraminone	2731.4	23.6183	286
Galanthamine-*N*-oxide	2638.4	22.5226	286
Epigalanthamine	2670.0	22.8942	287
Norlycoramine	2736.6	23.6786	274
**Ismine type**			
Ismine	2497.4	20.6170	238
**Other types of alkaloids**			
5,6-Dihydrolycorine	2384.9	21.6068	238
Demethylismine	2473.5	20.5799	224

**Table 5 life-15-00719-t005:** Evaluation of the AChE and BuChE inhibitory activities of Bolivian *Hippeastrum* alkaloid extracts. Their activity is expressed as the average of IC_50_ values (µg·mL^−1^) with standard deviations (SD). “NA” indicates no inhibitory activity for the specific enzyme in the corresponding species. Galanthamine (GAL) was included as a positive control.

		AChE		BuChE	
Code	Species	µg·mL^−1^ ± SD	Adjusted *p* Value	µg·mL^−1^ ± SD	Adjusted *p* Value
**H1**	*H. cardenasii*	49.67 ± 1.20	<0.0001	149.13 ± 3.02	<0.0001
**H2**	*H. chionedyanthum*	6.03 ± 0.45	0.3775	NA	-
**H4**	*H. cybister*	106.83 ± 4.15	<0.0001	NA	-
**H5**	*H. escobaruriae*	64.46 ± 4.50	<0.0001	NA	-
**H8**	*H. fragrantissimum*	11.85 ± 0.22	0.0032	103.62 ± 3.60	<0.0001
**H9**	*H. haywardii*	3.54 ± 0.09	0.9688	23.26 ± 0.24	0.0037
**H10**	*H. haywardii*	9.08 ± 0.64	0.0450	69.49 ± 2.73	<0.0001
**H12**	*H. lapacense*	75.07 ± 1.35	<0.0001	NA	-
**H13**	*H. lara-ricoi*	2.32 ± 0.07	0.9998	53.15 ± 2.44	<0.0001
**H14**	*H. leopoldii*	77.85 ± 3.38	<0.0001	NA	-
**H15**	*H. mollevillquense*	36.52 ± 2.85	<0.0001	NA	-
**H16**	*H. nelsonii*	51.66 ± 3.89	<0.0001	116.02 ± 5.00	<0.0001
**H18**	*H. pardinum*	102.90 ± 4.76	<0.0001	NA	-
**H19**	*H. parodii*	35.13 ± 1.75	<0.0001	119.23 ± 1.95	<0.0001
**H20**	*H. psittacinum*	94.45 ± 6.22	<0.0001	NA	-
**H21**	*H. puniceum*	141.46 ± 8.01	<0.0001	NA	-
**H23**	*H.* sp.	33.58 ± 1.18	<0.0001	111.70 ± 5.20	<0.0001
**H24**	*H. umabisanum*	162.10 ± 6.12	<0.0001	NA	-
**H25**	*H. vittatum*	36.68 ± 3.50	<0.0001	169.65 ± 7.53	<0.0001
**H26**	*H. warszewiczianum*	47.43 ± 1.79	<0.0001	NA	-
**H27**	*H. yungacense*	6.42 ± 0.21	0.3058	35.97 ± 0.81	<0.0001
**GAL**	**Galanthamine**	0.35 ± 0.07		3.58 ± 0.14	

## Data Availability

Data are contained within the article and in the Appendix A.

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
