# Peer review of "Alkaloid Profile Characterisation and Bioactivity Evaluation of Bolivian Hippeastrum Species (Amaryllidaceae) as Cholinesterase Inhibitors"

_life, 2025, doi:10.3390/life15050719_

Round 1

Reviewer 1 Report

Comments and Suggestions for Authors

The manuscript entitled “Alkaloid Profile Characterization and Bioactivity Evaluation of Bolivian Hippeastrum Species (Amaryllidaceae) as Cholinesterase Inhibitors” This manuscript presents a comprehensive phytochemical and pharmacological investigation of 27 Bolivian Hippeastrum species, identifying 48 Amaryllidaceae alkaloids and evaluating their acetylcholinesterase (AChE) and butyrylcholinesterase (BuChE) inhibitory activities. The study addresses a significant gap in the literature, as it is the first detailed characterization of most Bolivian Hippeastrum species, highlighting their potential for drug discovery, particularly for Alzheimer’s disease. The work is well-structured, methodologically sound, and contributes valuable data to the field of natural product research. However, minor revisions are required to improve clarity, address methodological limitations, and contextualize findings more effectively. Therefore, I just leave following comments for more clarification:

Major comments:

1- The use of relative quantification (based on galanthamine calibration) limits comparability with studies employing absolute quantification. The authors should explicitly acknowledge this limitation and discuss its implications for cross-study comparisons.

2- While in vitro cholinesterase inhibition data are robust, the lack of in vivo validation weakens translational relevance. A brief discussion on the need for future in vivo studies would strengthen the manuscript.

3- The discussion attributes bioactivity to individual alkaloids (e.g., lycorine, galanthamine), but synergistic interactions between compounds are not explored. This should be addressed as a potential factor influencing observed activities.

4- The conservation section in the introduction (lines 134–147) is tangential to the study’s primary focus. Consider condensing this and integrating ecological implications of alkaloid variability (e.g., defense mechanisms) into the discussion.

5- Data Presentation: Ensure Table S1 (alkaloid concentrations) is accessible and clearly formatted; Figure 3, The "Alkaloid-type characterization" figure lacks axis labels and a legend, making interpretation difficult. Revise for clarity; Table 3, Include units for IC50 values (µg·mL⁻¹) in the column headers for consistency.

6- The manuscript relies heavily on expert validation (e.g., Raúl Lara Rico). Include herbarium voucher numbers for all specimens in Table 1 to ensure reproducibility and compliance with taxonomic standards.

Minor comments:

7- Minor grammatical errors exist (e.g., "H27 and H29 present mostly lycorine" [H29 is not defined]). Revise for consistency in species codes. Avoid repetitive phrases (e.g., "notable quantity" appears frequently). Use synonyms or rephrase for variety.

8- In discussion section expand the ecological significance of alkaloid variability (e.g., adaptation to environmental stressors). Compare findings with prior studies on non-Bolivian Hippeastrum species to highlight uniqueness or commonalities.

9- Update citations to include recent reviews on Amaryllidaceae alkaloids (e.g., 2023 publications). Ensure all cited biological activities (e.g., lycorine’s anticancer effects) are supported by primary references.

Altogether the manuscript provides novel and impactful insights into the alkaloid diversity and bioactivity of Bolivian Hippeastrum species. Addressing the above points will enhance the manuscript’s rigor, clarity, and broader relevance.

Comments on the Quality of English Language

Minor grammatical errors exist

Author Response

Comment 1- The use of relative quantification (based on galanthamine calibration) limits comparability with studies employing absolute quantification. The authors should explicitly acknowledge this limitation and discuss its implications for cross-study comparisons.

Response 1.- An absolute quantification of each alkaloid would require a calibration curve for all que alkaloids, requiring a substantial amount of each individual compound. The relative quantification performed in this study is useful for measuring and comparing results between samples that have been extracted, purified and analysed using the same methodology. A sentence and a reference has been added in the Materials and Methods section of the manuscript clarifying this point.

Comment 2- While in vitro cholinesterase inhibition data are robust, the lack of in vivo validation weakens translational relevance. A brief discussion on the need for future in vivo studies would strengthen the manuscript.

Response 2.- A paragraph discussing this need has been included in the Discussion Section of the manuscript.

Comment 3- The discussion attributes bioactivity to individual alkaloids (e.g., lycorine, galanthamine), but synergistic interactions between compounds are not explored. This should be addressed as a potential factor influencing observed activities.

Response 3.- A paragraph commenting this factor has been included in the Discussion Section of the manuscript.

Comment 4- The conservation section in the introduction (lines 134–147) is tangential to the study’s primary focus. Consider condensing this and integrating ecological implications of alkaloid variability (e.g., defense mechanisms) into the discussion.

Response 4.- Tangential lines corresponding to conservation section have been delated.

Comment 5- Data Presentation: Ensure Table S1 (alkaloid concentrations) is accessible and clearly formatted; Figure 3, The "Alkaloid-type characterization" figure lacks axis labels and a legend, making interpretation difficult. Revise for clarity; Table 3, Include units for IC50 values (µg·mL⁻¹) in the column headers for consistency.

Response 5.- Legend is included in the pie chart. A clarification sentence with values information has been added in the figure legend. IC50 units have been added in Table 3 column headers.

Comment 6- The manuscript relies heavily on expert validation (e.g., Raúl Lara Rico). Include herbarium voucher numbers for all specimens in Table 1 to ensure reproducibility and compliance with taxonomic standards.

Response 6.- “Voucher number” has been corrected to “Collection Code” in the manuscript. The taxonomic identification of each specimen has been validated by a botanical specialist affiliated with a specific herbarium, as detailed in Table 2. Each specimen has been assigned a collection code, also provided in Table 2. At present, the specimens are undergoing the process of herborization in their respective herbaria. To enhance clarity and facilitate specimen tracking, we have included an additional column in Table 2 specifying the herbarium code where each specimen is currently housed.

Comment 7- Minor grammatical errors exist (e.g., "H27 and H29 present mostly lycorine" [H29 is not defined]). Revise for consistency in species codes. Avoid repetitive phrases (e.g., "notable quantity" appears frequently). Use synonyms or rephrase for variety.

Response 7.- H29 has been corrected to H9. Some changes have been performed to avoid word repetitions in Results section.

Comment 8- In discussion section expand the ecological significance of alkaloid variability (e.g., adaptation to environmental stressors). Compare findings with prior studies on non-Bolivian Hippeastrum species to highlight uniqueness or commonalities.

Response 8.- Ecological possible significance of alkaloids is mentioned in Results section. A paragraph has been included in Discussion section comparing findings with non-Bolivian Hippeastrum species.

Comment 9- Update citations to include recent reviews on Amaryllidaceae alkaloids (e.g., 2023 publications). Ensure all cited biological activities (e.g., lycorine’s anticancer effects) are supported by primary references.

Response 9.- References have been updated in Results section.

Altogether the manuscript provides novel and impactful insights into the alkaloid diversity and bioactivity of Bolivian Hippeastrum species. Addressing the above points will enhance the manuscript’s rigor, clarity, and broader relevance.

Reviewer 2 Report

Comments and Suggestions for Authors

The present study conducted a detailed phytochemical analysis of Bolivian species of Hippeastrum, identifying and quantifying 48 known alkaloids from this family, each with specific biological properties. Additionally, the study evaluated the inhibitory activity on acetylcholinesterase (AChE) and butyrylcholinesterase (BuChE), two key enzymes involved in neurodegenerative processes.

The "Materials and Methods" section is well-structured and detailed, providing information on the collection, identification, and compositional analysis of Hippeastrum species from Bolivia. Overall, the description is clear, methodical, and well-documented. The study offers a detailed inventory of the collected species, with precise geographical locations. The extraction parameters are clearly mentioned (temperature, solvents, pH, ultrasonication).

However, there are a few aspects that could be improved to enhance the scientific rigor of the study:

  1. A more detailed description of the ecological and climatic conditions of each collection site would be useful.
  2. It would be beneficial to specify the extraction yield (mg/g of dry material) for each species.
  3. Justifying the choice of method (why maceration and ultrasonication were chosen over other techniques such as Soxhlet extraction or microwave-assisted extraction) would improve methodological transparency.
  4. If there are specific references for the extraction method, they should be explicitly included.
  5. The tested extract concentration should be mentioned, along with whether a dose-response analysis was performed.
  6. I suggest specifying the number of replicates and including a statistical analysis of the results.
  7. It should be indicated whether enzyme stability under the testing conditions was verified.

Author Response

Comment 1.- A more detailed description of the ecological and climatic conditions of each collection site would be useful.

Response 1.- A new table (Table 1) has been added to de manuscript with detailed climatic and geographic characteristic of collection sites.

Comment 1.- It would be beneficial to specify the extraction yield (mg/g of dry material) for each species.

Response 2.- A table (new Table 2) has been added to the manuscript with the extraction yield.

Comment 3.- Justifying the choice of method (why maceration and ultrasonication were chosen over other techniques such as Soxhlet extraction or microwave-assisted extraction) would improve methodological transparency.

Response 3.- Amaryllidaceae alkaloids possess distinct polarity characteristics that determine their suitability for different extraction methods. Numerous studies have reported the use of the acid-base extraction method employed in the present study, as detailed in the Materials and Methods section. The objective of this research was not to compare different extraction methodologies; therefore, we selected the classical Amaryllidaceae alkaloid extraction method that has been previously reported to yield reliable results for Amaryllidaceae alkaloids. We appreciate the reviewer's comment and trust that this clarification adequately addresses the concern.

Comment 4. If there are specific references for the extraction method, they should be explicitly included.

Response 4.- Two references with the alkaloid extraction method used in this study are included in the Materials and Methods section of the manuscript.

Comment 5. The tested extract concentration should be mentioned, along with whether a dose-response analysis was performed.

Response 5.- A paragraph mentioning the primary dose-response analysis was added in the manuscript after the extract concentration range for each enzyme.

Comment 6. I suggest specifying the number of replicates and including a statistical analysis of the results.

Response 6.- A paragraph in the manuscript has been modified to provide clarification on the number of replicates and the statistical analysis in the Materials and Methods section. Two new columns have been added in new Table 5 with statistical analysis.

Comment 7. It should be indicated whether enzyme stability under the testing conditions was verified.

Response 7.- The acetyl- and butyrylcholinesterase assays have been successfully performed over many years, with numerous publications supporting this method, confirming that the enzymes remain stable and functional during analysis. Here we include some of the publications:

Ellman, G.L.; Courtney, K.D.; Andres, V.; Featherstone, R.M. A new and rapid colorimetric determination of acetylcholinesterase activity. Biochem. Pharmacol. 1961, 7, 88–90. https://doi.org/10.1016/0006-2952(61)90145-9

López, S.; Bastida, J.; Viladomat, F.; Codina, C. Acetylcholinesterase inhibitory activity of some Amaryllidaceae alkaloids and Narcissus extracts. Life Sci. 2002, 71:11, 2521–2529. https://doi.org/10.1016/S0024-3205(02)02034-9

Tallini, L.R.; Manfredini, G.; Rodríguez-Escobar, M.L.; Ríos, S.; Martínez-Francés, V.; Feresin, G.E.; Borges, W.d.S.; Bastida, J.; Viladomat, F.; Torras-Claveria, L. The Anti-Cholinesterase Potential of Fifteen Different Species of Narcissus L. (Amaryllidaceae) Collected in Spain. Life 2024, 14, 536. https://doi.org/10.3390/life14040536

Tallini, L.R.; Osorio, E.H.; Dos Santos, V.D.; Borges, W.S.; Kaiser, M.; Viladomat, F.; Zuanazzi, J.A.S.; Bastida, J. Hippeastrum reticulatum (Amaryllidaceae): Alkaloid profiling, biological activities and molecular docking. Molecules 2017, 22, 2191. https://doi.org/10.3390/molecules22122191

Torras-Claveria, L.; Berkov, S.; Codina, C.; Viladomat, F.; Bastida, J. Daffodils as potential crops of galanthamine. Assessment of more than 100 ornamental varieties for their alkaloid content and acetylcholinesterase inhibitory activity. Ind. Crops Prod. 2013, 43, 237-244. https://doi.org/10.1016/j.indcrop.2012.07.034

Reviewer 3 Report

Comments and Suggestions for Authors
  1. Please provide the yield of alkaloid extracts from plant samples.
  2. Please provide the AChE and BuChE inhibitory activities of 48 known Amaryllidaceae alkaloids.
  3. Please provide the selective comparison of Bolivian Hippeastrum alkaloid extracts for AChE and BuChE inhibition.
  4. Please provide experimental design and data on the competitiveness and selectivity of Bolivian Hippeastrum alkaloid extracts for AChE and BuChE inhibition.

Author Response

Comment 1. Please provide the yield of alkaloid extracts from plant samples.

Response 1.- The yield of alkaloid extracts has been included in new Table 3.

Comment 2. Please provide the AChE and BuChE inhibitory activities of 48 known Amaryllidaceae alkaloids.

Response 2.- The assessment of AChE and BuChE inhibition for each of the 48 identified alkaloids is beyond the scope of the present study. Isolating individual alkaloids is particularly challenging, as some are present in very low quantities and closely co-elute with structurally similar compounds, making purification difficult. However, we provide references on the AChE activity of some isolated Amaryllidaceae alkaloids:

López, S.; Bastida, J.; Viladomat, F.; Codina, C. Acetylcholinesterase inhibitory activity of some Amaryllidaceae alkaloids and Narcissus extracts. Life Sci. 2002, 71:11, 2521–2529. https://doi.org/10.1016/S0024-3205(02)02034-9

Berkov, S.; Atanasova, M.; Georgiev, B.; Bastida, J.; Doytchinova, I. The Amaryllidaceae alkaloids: an untapped source of acetylcholinesterase inhibitors. Phytochem. Rev. 202221, 1415–1443. https://doi.org/10.1007/s11101-021-09790-0

Comment 3. Please provide the selective comparison of Bolivian Hippeastrum alkaloid extracts for AChE and BuChE inhibition.

Response 3.- The selective comparison of values of AChE and BuChE obtained for Hippeastrum alkaloid extracts is included in Results Section. A discussion of possible synergisms is included in Discussion Section.

Comment 4. Please provide experimental design and data on the competitiveness and selectivity of Bolivian Hippeastrum alkaloid extracts for AChE and BuChE inhibition.

Response 4.- A paragraph discussing the future need of in vivo assays has been included in the Discussion Section of the manuscript. Another paragraph discussing synergism has also been included in the Discussion Section.

The acetyl- and butyrylcholinesterase assays have been successfully performed over many years, with numerous publications supporting this method. Here we include some of the publications:

Ellman, G.L.; Courtney, K.D.; Andres, V.; Featherstone, R.M. A new and rapid colorimetric determination of acetylcholinesterase activity. Biochem. Pharmacol. 1961, 7, 88–90. https://doi.org/10.1016/0006-2952(61)90145-9

López, S.; Bastida, J.; Viladomat, F.; Codina, C. Acetylcholinesterase inhibitory activity of some Amaryllidaceae alkaloids and Narcissus extracts. Life Sci. 2002, 71:11, 2521–2529. https://doi.org/10.1016/S0024-3205(02)02034-9

Tallini, L.R.; Manfredini, G.; Rodríguez-Escobar, M.L.; Ríos, S.; Martínez-Francés, V.; Feresin, G.E.; Borges, W.d.S.; Bastida, J.; Viladomat, F.; Torras-Claveria, L. The Anti-Cholinesterase Potential of Fifteen Different Species of Narcissus L. (Amaryllidaceae) Collected in Spain. Life 2024, 14, 536. https://doi.org/10.3390/life14040536

Tallini, L.R.; Osorio, E.H.; Dos Santos, V.D.; Borges, W.S.; Kaiser, M.; Viladomat, F.; Zuanazzi, J.A.S.; Bastida, J. Hippeastrum reticulatum (Amaryllidaceae): Alkaloid profiling, biological activities and molecular docking. Molecules 2017, 22, 2191. https://doi.org/10.3390/molecules22122191

Torras-Claveria, L.; Berkov, S.; Codina, C.; Viladomat, F.; Bastida, J. Daffodils as potential crops of galanthamine. Assessment of more than 100 ornamental varieties for their alkaloid content and acetylcholinesterase inhibitory activity. Ind. Crops Prod. 2013, 43, 237-244. https://doi.org/10.1016/j.indcrop.2012.07.034
